# Behavioral Correlates of COVID-19 Worry: Stigma, Knowledge, and News Source

**DOI:** 10.3390/ijerph182111436

**Published:** 2021-10-30

**Authors:** Gabriella Y. Meltzer, Virginia W. Chang, Sarah A. Lieff, Margaux M. Grivel, Lawrence H. Yang, Don C. Des Jarlais

**Affiliations:** 1Department of Social and Behavioral Sciences, School of Global Public Health, New York University, New York, NY 10003, USA; vc43@nyu.edu (V.W.C.); lieffs01@nyu.edu (S.A.L.); mg5757@nyu.edu (M.M.G.); ly1067@nyu.edu (L.H.Y.); 2Department of Population Health, Grossman School of Medicine, New York University, New York, NY 10003, USA; 3Department of Epidemiology, School of Global Public Health, New York University, New York, NY 10003, USA; don.desjarlais@nyu.edu; 4Department of Epidemiology, Mailman School of Public Health, Columbia University, New York, NY 10032, USA

**Keywords:** COVID-19, worry, stigma, news media, knowledge, depression, Health Belief Model

## Abstract

Non-adherence to COVID-19 guidelines may be attributable to low levels of worry. This study assessed whether endorsing COVID-19-stigmatizing restrictions, COVID-19 knowledge, and preferred news source were associated with being ‘very worried’ versus ‘not at all’ or ‘somewhat’ worried about contracting COVID-19. Survey data were collected in July–August 2020 from N = 547 New York State (NYS) and N = 504 national Amazon MTurk workers. Respondents who endorsed COVID-19 stigmatizing restrictions (NYS OR 1.96; 95% CI 1.31, 2.92; national OR 1.80; 95% CI 1.06, 3.08) and consumed commercial news (NYS OR 1.89; 95% CI 1.21, 2.96; national OR 1.93; 95% CI 1.24, 3.00) were more likely to be very worried. National respondents who consumed *The New York Times* (OR 1.52; 95% CI 1.00, 2.29) were more likely to be very worried, while those with little knowledge (OR 0.24; 95% CI 0.13, 0.43) were less likely to be very worried. NYS (OR 2.66; 95% CI 1.77, 4.00) and national (OR 3.17; 95% CI 1.95, 5.16) respondents with probable depression were also more likely to be very worried. These characteristics can help identify those requiring intervention to maximize perceived threat to COVID-19 and encourage uptake of protective behaviors while protecting psychological wellbeing.

## 1. Introduction

The outset of the COVID-19 pandemic caused by the novel SARS-CoV-2 coronavirus was characterized by a climate of widespread fear, worry, and uncertainty. Unlike other viral infections, over 50% of COVID-19 cases are contracted through presymptomatic and asymptomatic transmission, which further exacerbated levels of widespread panic and distrust of others [1]. As such, researchers have quickly mobilized to determine and communicate effective public health measures to mitigate the airborne spread of COVID-19, including social distancing, hand washing, limits on in-person (especially indoor) gatherings, mask wearing, and frequent testing. Several segments of the population, however, have refused to adhere to these guidelines, oftentimes belligerently [2,3]. Their choosing not to do so arises from a confluence of multi-level factors [4,5,6,7], but likely stems, at least in part, from their not being worried about contracting COVID-19.

The Health Belief Model is a widely used conceptual framework to explain the uptake of health behaviors such as those recommended to reduce the spread of COVID-19 [8]. The Health Belief Model posits several constructs that predict whether an individual will adopt disease preventive behaviors. The two constructs that are most salient to feelings of worry are perceived susceptibility and perceived severity. Perceived susceptibility is one’s “belief about the chances of experiencing a risk or getting a condition or disease,”, while perceived severity is their “belief about how serious a condition and its sequelae are [9].”. Perceived susceptibility and perceived severity together constitute perceived threat, which can be operationalized as an individual’s degree of worry about contracting the disease in question. These constructs are influenced by a complex interplay of sociodemographic, cultural, psychological, ideological, and structural variables [10,11,12,13] (see Figure 1). It is imperative that public health researchers and officials understand which of these factors contribute most to perceived threat or worry to effectively target interventions that could increase uptake of protective behaviors.

A growing body of international literature has found that various constructs of the Health Belief Model significantly predict the uptake of protective health behaviors such as handwashing, social distancing, and vaccination (previously hypothetical) to prevent the contraction and transmission of COVID-19. These include multinational studies [14], as well as studies conducted in India [15], Iran [16,17], Ethiopia [18,19], Korea [20], Egypt [21], Canada [22], and China [23]. One study in Italy specifically noted that those with greater perceived severity of COVID-19 in terms of incidence, mortality, social life, the economy, and health consequences showed high levels of worry [24]. In the U.S., social concern, perceived severity, and perceived barriers significantly predicted whether adolescents would seek a test for COVID-19 [25]. Guidry et al. (2021) found that perceived susceptibility, perceived benefits, and perceived barriers significantly predicted vaccine uptake in the U.S. under both normal and emergency use Food and Drug Administration authorization [26]. These studies highlight not only that perceived severity plays a strong role in predicting COVID-19 worry but that both perceived severity and susceptibility play a pivotal role in determining the uptake of frequent testing and vaccination, key behavioral components in reducing community spread of COVID-19.

Few studies to date, however, have examined the individual-level characteristics that predict a person’s degree of worry about contracting COVID-19 in the U.S. context, where large swaths of the population remain unwilling to adhere to effective public health directives [27,28,29]. These studies are limited though in that they do not address how worry may be related to stigma toward those unfairly associated with COVDI-19 or preferred news source. The objective of this study was therefore to determine those behavioral and sociodemographic factors associated with being ‘very worried’ about contracting COVID-19 as opposed to ‘not at all’ or ‘somewhat’ worried. Given the emotionally charged and highly politicized nature of the pandemic, along with the rampant spread of disinformation, we paid particular attention to how COVID-19 worry related to the endorsement of COVID-19 stigmatizing restrictions, preferred news source, and COVID-19 knowledge. We also assessed COVID-19 worry’s relationship to probable depression as an indicator of its potentially adverse relationship with psychological wellbeing. It is important to determine these behavioral and psychological correlates to frame further examination of COVID-19 worry as a precursor to either constructive or harmful behavior [30,31].

### 1.1. COVID-19 Stigma

Stigma is a process whereby individuals or groups are morally discredited and socially devalued based on a disease diagnosis or other trait [32]. Labeling occurs with social selection of a tag or designation to a person or group; these labels associate people with undesirable characteristics that can develop into stereotypes and lead to status loss and discrimination [33]. Throughout history, stigma has negatively impacted populations affected by diseases considered contagious, potentially deadly, and without a known cure [32], from plague, cholera, and yellow fever, to more recent diseases, such as HIV/AIDS, SARS, Ebola, Zika, and COVID-19. Health-related stigma is driven by fear of infection, misinformation, economic consequences of disease, lack of awareness, and socially constructed stereotypes [33,34,35,36,37]. Stigmatization leads to psychological, social, economic, and sometimes physical harm to those who are stigmatized with few discrete benefits of reducing disease transmission. It is, therefore, important to study the stigmatization of COVID-19 due to the very real possibility of these harms, which have materialized in discrimination and harassment toward those of Chinese descent. We therefore hypothesize that those who endorse COVID-19 stigmatizing restrictions will also exhibit greater worry about contracting the COVID-19 virus.

### 1.2. COVID-19 Knowledge

In a joint statement, the World Health Organization and other multinational agencies acknowledged the importance of technology and media platforms to increasing public knowledge by informing consumers about the latest developments in the COVID-19 pandemic [38]. Given that greater COVID-19 news consumption at the start of the pandemic was shown to be associated with anticipated mental health challenges (e.g., depression) [39], we hypothesize that those with greater knowledge of COVID-19 will express more worry about contracting the disease [40,41].

### 1.3. Preferred News Source

In addition to social media platforms such as Twitter and Facebook where it is difficult to regulate the spread of misinformation, conservative news platforms in the U.S. such as Fox News have advanced a narrative that downplays the seriousness of COVID-19 and discourages abiding by recommended public health guidelines [42,43,44]. On the other hand, studies have noted the overwhelming negative tone of stories by major U.S. media outlets and the relationship between exposure to the 24-h news cycle and poor psychological wellbeing [45,46]. We therefore hypothesize that those whose preferred news sources are social media and Fox News will express less COVID-19 worry, while consumers of traditional news outlets will express more.

### 1.4. Depression

Studies conducted throughout the world have consistently found that the lethal spread of COVID-19 and its associated lockdowns have been associated with greater levels of anxiety and depression at the population level [47,48,49,50,51,52,53,54,55,56,57,58,59,60,61]. Several of these studies also noted greater perceived risk of contracting COVID-19 to be significantly associated with depression [53,56,58,59,62]. We therefore hypothesize that similar to these other studies, respondents with probable depression will express greater worry of contracting COVID-19.

We assessed these dynamics throughout the U.S. as a whole and in New York State during the early months of the pandemic, when New York State was an epicenter and other parts of the country had yet to experience the worst of COVID-19 firsthand. New York State represents a test of those factors that are correlated with high levels of worry at the outset of a pandemic. The national sample, on the other hand, represents a test of generalizability of the New York State findings to determine if these same factors are associated with being very worried and if these relationships occur at the same magnitude in different U.S. locales.

## 2. Materials and Methods

Study data were collected as part of a larger project examining knowledge, beliefs, and attitudes toward HIV/AIDS, SARS, and COVID-19. Survey data were collected in July and August 2020 from 1219 Amazon MTurk workers through the Amazon MTurk service, an online crowd-sourcing platform that has successfully and effectively been used to launch online research surveys with fast response turnaround [63]. Respondents were eligible for the survey if they were 18 years or older, had previously completed at least 500 Human Intelligence Tasks, and had Human Intelligence Task approval ratings of least 90%. Participants were initially required to be residents of New York State, where state residence was determined through the software’s demographic filtering feature. Residence eligibility was later expanded to the entire U.S. (See Appendix A for detailed information on the study phases, including dates and phase-specific inclusion criteria). Detailed information on data collection, selection criteria, measures, and data quality control can be found elsewhere [64].

Study participants were assessed regarding their degree of worry about contracting COVID-19, knowledge of the disease, and their beliefs and attitudes towards hypothetical methods of controlling its spread to discern underlying stigmatizing attitudes toward high-risk groups, namely individuals of Chinese descent. Respondents were also queried on their demographic characteristics, preferred news source, and probable current depression. Survey items assessing COVID-19 worry, knowledge and stigma were modified from previous work on the stigmatization of HIV/AIDS and SARS and their associated high-risk groups (those who identify as gay and those of Chinese descent, respectively) among New York City residents in the aftermath of the September 11, 2001 terrorist attacks [65].

Survey participants were compensated for roughly ten minutes of their time and provided online informed consent. All study procedures were approved by the New York University Institutional Review Board.

### Statistics

The final analytical sample sizes for this analysis were N = 547 for the New York State sample and N = 504 for the national sample. Respondents who failed at least one attention check (*n* = 35), had inadequate survey completion times (*n* = 65), and/or who reported that they were entirely unaware of COVID-19 (*n* = 53) were removed from the analysis. The New York State and national samples were analyzed separately due to the oversampling of respondents from New York State, unique sample characteristics, and regional differences in severity of the pandemic at the time of survey distribution.

The primary outcome of interest was a respondent’s degree of worry about contracting COVID-19, characterized as ‘not at all’, ‘somewhat’, or ‘very’ worried. COVID-19 stigmatizing restrictions as a behavioral exposure was determined based on ‘somewhat’ or ‘strongly’ agreeing with the following statements adapted from the original HIV/AIDS and SARS study: (1) requiring Americans with COVID-19 to wear identification tags; (2) the government announcing it will execute people who knowingly spread COVID-19; (3) avoiding areas of the U.S. heavily populated by Chinese individuals; (4) forcing all Chinese people to be medically checked for COVID-19; and (5) not allowing Chinese people to enter the U.S. [65]. COVID-19 stigma was tested as a binary variable, where individuals either held no stigmatizing beliefs (ref.) or agreed with at least one of the aforementioned statements. COVID-19 knowledge was based on respondents’ self-reported response to the question, “How much have you heard about COVID-19?” with options being ‘a great deal,’ ‘some,’ or ‘not much.’ For purposes of this analysis, COVID-19 knowledge was dichotomized as a great deal (ref.) versus some or not much. Respondents were also asked, “From which of these sources do you regularly get your news?” Preferred news source as an additional behavioral exposure was based on dummy variables for the following categories: commercial news (ABC, NBC, CNN, MSNBC, and CBS); *The New York Times*; social media (Facebook and Twitter), publicly funded news (PBS and NPR), and Fox News.

Sociodemographic exposures of interest included sex (male [ref.] vs. female); racial/ethnic self-identification (Non-Hispanic White [ref.], Non-Hispanic Black, Hispanic, or Asian); age group (18–24 [ref.], 25–34, 35–44, 45–54, 55+); education level (high school degree [ref.] vs. college degree or greater), and employment status (unemployed [ref.], full-time employment, part-time employment). Probable current depression (none [ref.] vs. probable) was based on the widely validated Patient Health Questionnaire, where probable depression was based on an established cut-off score of five or greater [66].

We calculated univariate and bivariate statistics followed by logistic regression to evaluate the factors significantly associated with COVID-19 worry for the New York State and national samples. Bivariate analyses examined COVID-19 worry based on its three original categories of ‘not at all’, ‘somewhat’, or ‘very’ worried. Following the Health Belief Model’s presumption that those who were ‘very worried’ would be most likely to take preventive health measures and adhere to public health guidelines [67], we dichotomized COVID-19 into ‘not at all or somewhat’ (ref.) versus ‘very’ worried for multivariable analyses. All analyses were conducted using STATA 15 [68].

## 3. Results

Table 1 shows the results of univariate and bivariate analyses for the New York State sample. Most respondents were female (54.7%), had at least a college degree (67.3%), were employed full-time (60.5%), and largely identified as Non-Hispanic White (68.6%), followed by Asian (11.0%), Hispanic (10.4%), and Non-Hispanic Black (9.1%). Roughly 50% of respondents were aged 35 or older, and over half (54.6%) had probable depression. Forty percent endorsed COVID-19 stigmatizing restrictions and 95.1% had ‘a great deal’ of knowledge about the disease. The most popular preferred news sources included commercial news (69.1%), followed by social media (54.4%), *The New York Times* (48.4%), public funded news (43.2%), and Fox News (22.2%). Bivariate analyses showed that those who endorsed COVID-19 stigmatizing restrictions (*p* < 0.001), those with probable depression (*p* < 0.001), and those who consumed commercial news (*p* < 0.001), and *The New York Times* (*p* = 0.001) tended to be more worried about contracting COVID-19. Watching Fox News was associated with less worry (*p* = 0.016).

Table 2 shows the results of univariate and bivariate analyses for the national sample. Unlike the New York State sample, most respondents were male (63.9%) and over half were under the age of 34. Although the majority identified as Non-Hispanic White (57.1%), greater proportions identified as Hispanic (26.4%) and Non-Hispanic Black (13.5%), while only 3% identified as Asian. Larger majorities of the national sample had a college degree or greater (74.8%), full-time employment (79.8%), and probable depression (65.7%). Other marked differences include that in the national sample, the majority (65.7%) of respondents endorsed COVID-19 stigma and a greater proportion had ‘some’ or ‘not much’ knowledge of COVID-19 (16.9%). The consumption of news sources also differed slightly, with 73.0% using social media, 28.6% public funded news, and 39% Fox News. Bivariate analyses showed that those who endorsed COVID-19 stigmatizing restrictions (*p* < 0.001), identified as Hispanic (*p* = 0.026), had a college degree or greater (*p* < 0.001), were employed full-time (*p* = 0.005), had probable depression (*p* < 0.001), had a great deal of knowledge about COVID-19 (*p* < 0.001), and who consumed commercial news (*p* < 0.001), *The New York Times* (*p* < 0.001), and social media (*p* = 0.035) tended to be more worried about contracting COVID-19.

Table 3 shows the results of logistic regression predicting being ‘very’ worried about contracting COVID-19 versus ‘not at all or somewhat’ worried in the New York State sample controlling for sex, race/ethnicity, age, education level, employment status, probable depression, COVID-19 stigmatizing restrictions, preferred news source, and COVID-19 knowledge. New York State respondents who endorsed COVID-19 stigmatizing restrictions were significantly more likely than their non-stigmatizing counterparts to be ‘very’ worried about contracting COVID-19 (OR 1.96; 95% CI 1.31, 2.92). Those who consumed commercial news were significantly more likely than those who did not to express being ‘very’ worried about contracting COVID-19 (OR 1.89; 95% CI 1.21, 2.96). In addition, those with probable depression had 2.66 times greater odds (95% CI 1.77, 4.00) of being ‘very’ worried about contracting COVID-19 as compared to those without.

Table 4 shows the result of the multivariate analysis predicting being ‘very’ worried about contracting COVID-19 in the national sample, controlling for the same variables. Similar to the New York State sample, those who endorsed COVID-19 stigmatizing restrictions were significantly more likely to be ‘very’ worried about contracting COVID-19 (OR 1.80; 95% CI 1.06, 3.08) as compared to those who did not. National respondents who consumed commercial news (OR 1.93; 95% CI 1.24, 3.00) and *The New York Times* (OR 1.52; 95% CI 1.00, 2.29) were significantly more likely to express being ‘very’ worried about contracting COVID-19 as compared to those who did not prefer these news sources. On the other hand, those with ‘some’ or ‘not much’ knowledge (OR 0.24; 95% CI 0.13, 0.43) were significantly less likely than those with ‘a great deal’ of knowledge to be ‘very’ worried about contracting COVID-19. Those with probable depression were also more likely to be ‘very’ worried than those without (OR 3.17; 95% CI 1.95, 5.16).

## 4. Discussion

This study examined the behavioral and sociodemographic factors associated with being very worried about contracting COVID-19 among samples of U.S. adults from New York State and nationally prior to vaccine availability, when transmission was entirely dependent on adherence to public health measures. Considering the political divisiveness and the normalization of disinformation throughout the pandemic, we focused on whether the endorsement of COVID-19 stigmatizing restrictions, preferred news source, and COVID-19 knowledge were related to whether respondents expressed being very worried about contracting the virus. Of note was the substantial overall prevalence of worry in both samples; we found that 36% of New York State respondents and 45% of national respondents were very worried about COVID-19. In New York State, those who were more likely to endorse stigmatizing restrictions, consume commercial news, and have probable depression were significantly more likely to be very worried about contracting COVID-19. In the national sample, those who were very worried about contracting COVID-19 were significantly more likely to endorse stigmatizing restrictions, consume commercial news and *The New York Times*, and have probable depression. In addition, those with little knowledge about the pandemic were less likely to be very worried about contracting the virus. These findings may reflect the early phase of the pandemic during which most people did not have a clear understanding of COVID-19 and the federal government failed to use an evidence-based approach in its response [69]. The fact that these two sets of findings from New York State and the national sample are similar shows that the factors associated with being very worried about contracting COVID-19 are replicable and potentially generalizable from the single state to a nationally based sample, at least in the U.S.

The association between endorsement of COVID-19 stigmatizing restrictions and worry about contracting the disease is consistent with recent studies showing the negative impact of COVID-19 stigma on taking recommended public health actions, and the notion that stigmatization and fear of infection are closely related. In a qualitative study of Finland households with at least one confirmed case of COVID-19, perceived stigma among respondents was a product of fear and blame for infection, which manifested in reticence to disclose their COVID-19 status to others [70]. Similarly, a study in Malawi found that those who perceived stigma associated with COVID-19 were less likely to seek a test for the virus [71]. These findings from outside the U.S. appear to operate in contrast to a simple relationship between COVID-19 stigma, worry, and the uptake of protective health behaviors. Further research is needed to better understand how stigma influences adherence to recommended public health measures to reduce the spread of COVID-19.

The significant relationship between COVID-19 stigma and worry demonstrates the limitations of relying on the fear-driven components of the Health Belief Model (i.e., perceived threat) as a conceptual tool to reduce COVID-19 transmission. As used by public health educators, the Health Belief Model is fundamentally a “rational” behavior model. An individual perceives a threat to his or her health, identifies an action that that will remove (or at least mitigate) the threat, performs the action, and is relieved that the threat has been removed (or at least ameliorated). Our analyses reveal that being very worried about COVID-19 is associated with a negative response of endorsing stigmatizing attitudes and behavioral restrictions. However, the attitudes and restrictions contained in this five-item scale would not be practical methods for controlling the transmission of SARS-CoV-2, as they promote policies that are not only irrational, but harmful to groups unfairly associated with the virus. This is exemplified by a dramatic increase in violence and harassment toward Asians and Asian Americans in the U.S., with over 9000 incidents reported to Stop AAPI Hate from March 2020 to June 2021 [72]. These troubling dynamics reveal the difficulties in using fear arousal to address transmission of an infectious disease that is associated with an already stigmatized social group. Further research is needed to illuminate how to increase the type of concern that leads to the positive uptake of protective behaviors without increasing stigmatization and harmful conduct toward those who have contracted or are wrongly associated with COVID-19 or other infectious diseases.

The observed relationship between knowledge of COVID-19 and worry about contracting the virus is also in agreement with past literature on other infectious diseases. Studies on H1N1, HIV, HPV, and tuberculosis have shown that those with greater knowledge of the disease tend to exhibit greater worry, and vice versa [73,74,75,76]. Applying this to the Health Belief Model, those with greater knowledge of SARS-CoV-2′s virulence and lethality would likely perceive themselves as more susceptible and the virus as a severe hazard, therefore embodying greater perceived threat.

Our findings also demonstrate the importance of the association between preferred news source and levels of COVID-19 worry. Those respondents who consumed commercial news in both the New York State and national samples, and *The New York Times* in the national sample, were more likely to express greater degrees of worry. This is consistent with a study from early in the pandemic demonstrating a significant positive correlation between greater consumption of COVID-19-related cable news and perceived threat to both population health and the economy [77]. Although national consumers of *The New York Times*, a liberal-learning newspaper, may not yet have had firsthand exposure to COVID-19 at this point in the pandemic, they absorbed the images from places such as New York that were severely impacted and read in-depth analyses published by experts warning of the virus’s potential impact. In fact, one study showed that rural residents whose news was produced in a city more impacted by COVID-19 were more likely to engage in social distancing than otherwise similar rural residents [78].

It is also not surprising that those with probable depression were also more likely to express high degrees of worry about contracting COVID-19, as psychological research has solidified the close relationship between depression, anxiety, and symptoms of worry across diverse populations [79,80,81]. It is, therefore, important that public health interventions be aimed toward increasing concern rather than worry about infection so as not to inadvertently harm collective psychological wellbeing (e.g., increase depression). This is especially the case for certain subpopulations that have disproportionately experienced poor mental health and reduced quality of life as a result of the COVID-19 pandemic, including students, pregnant women, frontline healthcare workers, and those of East Asian descent who have experienced verbal and physical harassment [82,83,84,85].

Several studies, on the other hand, identified sociodemographic factors associated with COVID-19 worry that were either not significant or not assessed in the context of this analysis. These include older age, male gender, socioeconomic status, sense of community, and living with larger families [29,86,87]. The sociodemographic factors shown to be significant in our study may not have showed differences in these other studies due to differing country contexts.

### Limitations

This study has several limitations. Given its cross-sectional nature, it is impossible to infer whether there is, in fact, a causal relationship between the explored sociodemographic and behavioral factors and an individual’s level of worry about contracting COVID-19; this leaves concern for reverse causality. There is also the potential for circular causation in that respondents prefer those media sources that are consistent with their prior beliefs, and consuming those sources only strengthens their preconceived perceptions and attitudes surrounding COVID-19. In addition, the nature of respondent recruitment using Amazon MTurk limits the representativeness of the sample to those with Internet access who use this service to earn supplemental income. The respondents in our study had higher education levels than the general population, and it was also not possible to discern whether they came from urban or rural areas, which may have also influenced their responses. This study was also subject to social desirability bias in that respondents may not have wanted to disclose their true levels of stigmatizing attitudes.

## 5. Conclusions

This analysis is one of a handful of studies that assesses the sociodemographic and behavioral factors associated with an individual’s level of worry about contracting COVID-19 in the early months of the pandemic prior to the availability of highly effective vaccines. Those characteristics that are significantly associated with high levels of COVID-19 worry—specifically the endorsement of COVID-19 stigmatizing restrictions, greater COVID-19 knowledge, and the consumption of certain news outlets—can be used to identify individuals and communities requiring greater, more innovative interventions to maximize perceived threat to the virus to encourage the uptake of health behaviors to protect themselves and others, while minimizing any potential negative psychological responses, i.e., while greater education by public health officials about COVID-19 could lead to greater levels of worry and wider uptake of preventive behaviors, there also needs to be a realization of potential negative effects such as increased levels of stigma and poor psychological wellbeing (i.e., depression). Future research is needed to understand how additional constructs of the Health Belief Model not examined by our analysis, including perceived benefits, perceived barriers, self-efficacy, and cues to action, predict how various subsets of the population will respond to shifting public health guidelines, especially vaccination, as the COVID-19 pandemic continues to unfold.

## Figures and Tables

**Figure 1 ijerph-18-11436-f001:**
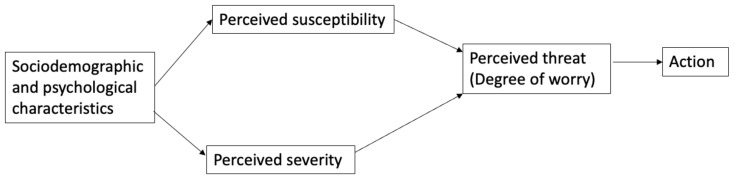
Adapted Health Belief Model.

**Table 1 ijerph-18-11436-t001:** Associations between MTurk worker characteristics and COVID-19 worry, New York State, N = 547.

	Total	COVID-19 Worry			
	N (%) N = 547	Not at All *n* (%) *n* = 74	Somewhat *n* (%) *n* = 274	Very *n* (%) *n* = 199	*p-*Value ^1^
**COVID-19 Stigma**					
No	329 (60.15)	46 (62.16)	185 (67.25)	98 (49.25)	
Yes	218 (39.85)	28 (37.84)	89 (32.48)	101 (50.75)	<0.001
**Race/Ethnicity**					
Non-Hispanic White	375 (68.56)	51 (68.92)	193 (70.44)	131 (65.83)	
Non-Hispanic Black	55 (9.05)	10 (13.51)	27 (9.85)	18 (9.05)	
Hispanic	57 (10.42)	9 (12.16)	24 (8.76)	24 (12.06)	0.447
Asian	60 (10.97)	4 (5.41)	30 (10.95)	26 (13.07)	
**Sex ^2^**					
Male	244 (45.27)	35 (47.95)	121 (44.98)	88 (44.67)	
Female	295 (54.73)	38 (52.05)	148 (55.02)	109 (55.33)	0.883
**Age**					
18–24	52 (9.51)	4 (5.41)	29 (10.58)	19 (9.55)	
25–34	217 (39.67)	34 (45.95)	98 (35.77)	85 (42.71)	
35–44	148 (27.06)	16 (21.62)	82 (29.93)	50 (25.13)	0.101
45–54	69 (12.61)	12 (16.22)	27 (9.85)	30 (15.08)	
55+	61 (11.15)	8 (10.81)	38 (13.87)	15 (7.54)	
**Education Level**					
High school	179 (32.72)	33 (44.59)	87 (31.75)	59 (29.65)	
College degree or greater	368 (67.28)	41 (55.41)	187 (68.25)	140 (70.35)	0.058
**Employment Status**					
Not Employed	117 (21.39)	17 (22.97)	58 (21.17)	42 (21.11)	
Part-time Employment	99 (18.10)	8 (10.81)	51 (18.61)	40 (20.10)	0.517
Full-time Employment	331 (60.51)	49 (66.22)	165 (60.22)	117 (58.79)	
**Probable Depression** ^2^					
No Probable Depression	247 (45.40)	46 (62.16)	142 (52.40)	59 (29.65)	
Probable Depression	297 (54.60)	28 (37.84)	129 (47.60)	140 (70.35)	<0.001
**News Source**					
Commercial News ^2^					
No	169 (30.95)	36 (48.65)	90 (32.97)	43 (21.61)	
Yes	377 (69.05)	38 (51.35)	183 (67.03)	156 (78.39)	<0.001
New York Times ^2^					
No	282 (51.65)	52 (70.27)	139 (50.92)	91 (45.73)	
Yes	264 (48.35)	22 (29.73)	134 (49.08)	108 (54.27)	0.001
Social Media ^2^					
No	249 (45.60)	36 (48.65)	124 (45.42)	89 (44.72)	
Yes	297 (54.40)	38 (51.35)	149 (54.58)	110 (55.28)	0.843
Publicly Funded ^2^					
No	310 (56.78)	42 (56.76)	150 (54.95)	118 (59.30)	
Yes	236 (43.22)	32 (43.24)	123 (45.05)	81 (40.70)	0.641
Fox News ^2^					
No	425 (77.84)	49 (66.22)	223 (81.68)	153 (76.88)	
Yes	121 (22.16)	25 (33.78)	50 (18.32)	46 (23.12)	0.016
**COVID-19 Knowledge**					
A great deal	520 (95.06)	69 (93.24)	259 (94.53)	192 (96.48)	
Some/not much	27 (4.94)	5 (6.76)	15 (5.47)	7 (3.52)	0.462

^1^ Pearson’s chi-square unless otherwise noted. ^2^ Totals do not sum to N due to missing values.

**Table 2 ijerph-18-11436-t002:** Associations between MTurk worker characteristics and COVID-19 worry, National, N = 504.

	Total	COVID-19 Worry			
	N (%) N = 504	Not at All *n* (%) *n* = 57	Somewhat *n* (%) *n* = 219	Very *n* (%) *n* = 228	*p-*Value ^1^
**COVID-19 Stigma**					
No	173 (34.33)	28 (49.12)	91 (41.55)	54 (23.68)	
Yes	331 (65.67)	29 (50.88)	128 (58.45)	174 (76.32)	<0.001
**Race/Ethnicity**					
Non-Hispanic White	288 (57.14)	41 (71.93)	131 (59.82)	116 (50.88)	
Non-Hispanic Black	68 (13.49)	6 (10.53)	31 (14.16)	31 (13.60)	
Hispanic	133 (26.39)	7 (12.28)	51 (23.29)	75 (32.89)	0.026
Asian	15 (2.98)	3 (5.26)	6 (2.74)	6 (2.63)	
**Sex** ^2^					
Male	321 (63.94)	36 (63.16)	142 (65.44)	143 (62.72)	
Female	181 (36.06)	21 (36.84)	75 (34.56)	85 (37.28)	0.830
**Age**					
18–24	26 (5.16)	5 (8.77)	7 (3.20)	14 (6.14)	
25–34	233 (46.23)	19 (33.33)	100 (45.66)	114 (50.00)	
35–44	147 (29.17)	17 (29.82)	69 (31.51)	61 (26.75)	0.181
45–54	54 (10.71)	10 (17.54)	21 (9.59)	23 (10.09)	
55+	44 (8.73)	6 (10.53)	22 (10.05)	16 (7.02)	
**Education Level**					
High school	127 (25.20)	26 (45.61)	59 (26.94)	42 (18.42)	
College degree or greater	377 (74.80)	31 (54.39)	160 (73.06)	186 (81.58)	<0.001
**Employment Status**					
Not Employed	43 (8.53)	11 (19.30)	19 (8.68)	13 (5.70)	
Part-time Employment	59 (11.71)	9 (15.79)	29 (13.24)	21 (9.21)	0.005
Full-time Employment	402 (79.76)	37 (64.91)	171 (78.08)	194 (85.09)	
**Probable Depression**					
No Probable Depression	173 (34.33)	36 (63.13)	93 (42.47)	44 (19.30)	
Probable Depression	331 (65.67)	21 (36.84)	126 (57.53)	184 (80.70)	<0.001
**News Source**					
Commercial News					
No	167 (33.13)	37 (64.91)	73 (33.33)	57 (25.00)	
Yes	337 (66.87)	20 (35.09)	146 (66.67)	171 (75.00)	<0.001
New York Times					
No	303 (60.12)	46 (80.70)	140 (63.93)	117 (51.32)	
Yes	201 (39.88)	11 (19.30)	79 (36.07)	111 (48.68)	<0.001
Social Media					
No	136 (26.98)	22 (38.60)	63 (28.77)	51 (22.37)	
Yes	368 (73.02)	35 (61.40)	156 (71.23)	177 (77.63)	0.035
Publicly Funded					
No	360 (71.43)	35 (61.40)	156 (71.23)	169 (74.12)	
Yes	144 (28.57)	22 (38.60)	63 (28.77)	59 (25.88)	0.163
Fox News					
No	309 (61.31)	41 (71.93)	138 (63.01)	130 (57.02)	
Yes	195 (38.69)	16 (28.07)	81 (36.99)	98 (42.98)	0.093
**COVID-19 Knowledge** ^2^					
A great deal	418 (83.10)	47 (82.46)	165 (75.69)	206 (90.35)	
Some/not much	85 (16.90)	10 (17.54)	53 (24.31)	22 (9.65)	<0.001

^1^ Pearson’s chi-square unless otherwise noted. ^2^ Totals do not sum to N due to missing values.

**Table 3 ijerph-18-11436-t003:** Logistic regression predicting being very COVID-19 worried, New York State, N = 535.

	Crude Odds Ratios (95% CI)	Multivariate Odds Ratio (95% CI)
**COVID-19 Stigma**		
No	1.00	1.00
Yes	2.03 (1.43, 2.90) ***	1.96 (1.31, 2.92) **
**Race/Ethnicity**		
Non-Hispanic White	1.00	1.00
Non-Hispanic Black	0.91 (0.50, 1.65)	0.94 (0.49, 1.81)
Hispanic	1.35 (0.77, 2.39)	1.20 (0.64, 2.26)
Asian	1.42 (0.82, 2.48)	1.31 (0.70, 2.43)
**Sex**		
Male	1.00	1.00
Female	1.04 (0.73, 1.48)	0.98 (0.66, 1.46)
**Age**		
18–24	1.00	1.00
25–34	1.12 (0.60, 2.09)	1.40 (0.70, 2.79)
35–44	0.89 (0.46, 1.71)	1.13 (0.54, 2.35)
45–54	1.34 (0.64, 2.80)	2.09 (0.91, 4.82)
55+	0.57 (0.25, 1.27)	0.81 (0.33, 1.98)
**Education Level**		
High school degree	1.00	1.00
College degree or greater	1.25 (0.86, 1.82)	1.14 (0.74, 1.75)
**Employment Status**		
Not employed	1.00	1.00
Part-time employment	1.21 (0.70, 2.10)	1.29 (0.71, 2.38)
Full-time employment	0.98 (0.63, 1.52)	1.00 (0.60, 1.68)
**Probable Depression**		
No probable depression	1.00	1.00
Probable depression	2.84 (1.96, 4.12) ***	2.66 (1.77, 4.00) ***
**News Source**		
Commercial news	2.07 (1.38, 3.09) ***	1.89 (1.21, 2.96) **
New York Times	1.45 (1.02, 2.06) *	1.20 (0.81, 1.80)
Social media	1.06 (0.75, 1.50)	0.80 (0.54, 1.19)
Publicly funded	0.85 (0.60, 1.21)	0.98 (0.65, 1.48)
Fox News	1.09 (0.72, 1.65)	0.97 (0.61, 1.55)
**COVID-19 Knowledge**		
A great deal	1.00	1.00
Some/not much	0.60 (0.25, 1.44)	0.53 (0.20, 1.36)

** p* < 0.05, ** *p* < 0.01, *** *p* < 0.001.

**Table 4 ijerph-18-11436-t004:** Logistic regression predicting being very COVID-19 worried, National, N = 501.

	Crude Odds Ratios (95% CI)	Multivariate Odds Ratio (95% CI)
**COVID-19 Stigma**		
No	1.00	1.00
Yes	2.44 (0.66, 3.60) ***	1.80 (1.06, 3.08) *
**Race/Ethnicity**		
Non-Hispanic White	1.00	1.00
Non-Hispanic Black	1.24 (0.73, 2.12)	0.93 (0.51, 1.72)
Hispanic	1.92 (1.27, 2.91) **	1.41 (0.85, 2.36)
Asian	0.99 (0.34, 2.85)	1.34 (0.42, 4.26)
**Sex**		
Male	1.00	1.00
Female	1.10 (0.76, 1.59)	1.13 (0.74, 1.72)
**Age**		
18–24	1.00	1.00
25–34	0.82 (0.36, 1.85)	0.87 (0.35, 2.21)
35–44	0.61 (0.26, 1.41)	0.83 (0.32, 2.18)
45–54	0.64 (0.25, 1.63)	0.95 (0.33, 2.78)
55+	0.49 (0.18, 1.31)	0.79 (0.26, 2.39)
**Education Level**		
High school degree	1.00	1.00
College degree or greater	1.97 (1.29, 3.00) **	1.21 (0.73, 2.00)
**Employment Status**		
Not employed	1.00	1.00
Part-time employment	1.28 (0.55, 2.96)	0.85 (0.33, 2.18)
Full-time employment	2.15 (1.09, 4.25) *	1.43 (0.65, 3.14)
**Probable Depression**		
No probable depression	1.00	1.00
Probable depression	3.67 (2.45, 5.50) ***	3.17 (1.95, 5.16) ***
**News Source**		
Commercial news	1.99 (1.35, 2.92) ***	1.93 (1.24, 3.00) **
New York Times	1.96 (0.37, 2.81) ***	1.52 (1.00, 2.29) *
Social media	1.54 (1.03, 2.31) *	1.13 (0.70, 1.83)
Publicly funded	0.78 (0.53, 1.16)	1.20 (0.73, 1.96)
Fox News	1.39 (0.97, 1.99)	1.03 (0.66, 1.59)
**COVID-19 Knowledge**		
A great deal	1.00	1.00
Some/not much	0.36 (0.21, 0.61) ***	0.24 (0.13, 0.43) ***

* *p* < 0.05, ** *p* < 0.01, *** *p* < 0.001.

## Data Availability

The data presented in this study are available upon request from the corresponding and senior author.

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
