# Peer review of "Behavioral Correlates of COVID-19 Worry: Stigma, Knowledge, and News Source"

_ijerph, 2021, doi:10.3390/ijerph182111436_

Round 1

Reviewer 1 Report

Review of “Behavioral correlates of COVID-19 worry:

stigma, knowledge, and news source

(Manuscript ID: ijerph-1435765)

General comments: Generally, this study tended to answer two important questions: 1) The relationship between behavioral of COVID-19 worry and COVID-19 stigmatizing restrictions, COVID-19 knowledge, news source, depression. 2) Clarify the regional differences of behavioral of COVID-19 worry. The New York State and national samples were analyzed separately due to regional differences in severity of the pandemic at the time of survey distribution. These characteristics can help identify those requiring intervention to maximize perceived threat to COVID-19 and encourage uptake of protective behaviors while protecting psychological wellbeing, contributes to effective governance and policy implementation of COVID-19 prevention and control. Nevertheless, several concerns should be addressed properly before the paper can be published in the International Journal of Environmental Research and Public Health.

First, in general, the description of the Health Belief Model was unclear, please make efforts to revise this section. For instance, the authors proposed that “The Health Belief Model is a widely used conceptual framework to explain the uptake of health behaviors……” (Line 38-39, Page 1), the description of the Health Belief Model was not sufficient, and more literature citations are necessary. In addition, the rationality of the definition of the concepts of perceived susceptibility, perceived severity and perceived threat needs to be supplemented (“Perceived susceptibility is the likelihood that……in question.” Line 42-47, Page 1-2). Thirdly, perceived benefits, perceived barriers and self-efficacy seems to be redundant (Figure1, Page 2), have no relevant to the study and need to be revised. Furthermore, based on the Health Belief Model, how behavioral of COVID-19 worry, COVID-19 stigmatizing restrictions, COVID-19 knowledge, news source and depression fit this Model? Please give further explanation.

Second, the value and significance of this study was not clear, the conclusions of the study were obvious, new addition to the knowledge base needs to be clear. As the findings stand, little new information is graspable. That is, the impacts of the COVID-19 stigmatizing restrictions, COVID-19 knowledge, news source and depression on behavioral of COVID-19 worry are too obvious. There is simply no surprise with the findings. Great crafting is necessary to articulate the value of work and pinpoint new knowledge created.

Third, in general, the rationale for conducting this study was unclear, please make efforts to revise the introduction section. And in the literature, please provide more evidence to address the purpose of this study is critical.

Fourth, there are few explanations of the rationality for the materials and methods. For instance, how “COVID-19 Knowledge” is measured was not mentioned in the study, “COVID-19 Knowledge” was divided into “A great deal” and “Some/not much” (Table1, Page 5-6). What kind of respondents were defined as "A great deal" and what kind of respondents were defined as "Some/not much", which needs to be supplemented. In addition, how “Depression” was measured? “Depression” was divided into “No Probable Depression” and “Probable Depression” (Table1, Page 5-6), it should be described in detail. Furthermore, the authors were asked to clarify the causal relationship between “depression” and “worry of contracting COVID-19”, it is very necessary. For example, there are contradictory statements in the study. “Several of these studies also noted greater perceived risk of contracting COVID-19 to be a significant predictor of depression” (Line 120-122, Page 3). This sentence shows that “perceived risk of contracting COVID-19” affect “depression”. “We therefore hypothesize that, similar to these other studies, respondents with probable depression will express greater worry of contracting COVID-19” (Line 122-123, Page 3). However, this sentence shows that “depression” affect “worry of contracting COVID-19”.

Last but not least, it is noted that the manuscript needs careful editing by someone with expertise in technical English editing paying particular attention to English grammar, spelling, and sentence structure so that the goals and results of the study are clear to the reader.

Specific comments:

1.Line 96-106, Page 3: What does “COVID-19 Knowledge” mean in the study?

The authors were asked to define what “COVID-19 Knowledge” is.

2.Line 96-106, Page 3: In this section, the author needs to explain the relationship between “COVID-19 Knowledge” and “behavior of COVID-19 worry”, however, the authors was emphasizing the role of social media in public health safety (Line 98-100, Page 3: “the importance of technology and social media in keeping the public informed about the latest developments in the COVID-19 pandemic and updated guidelines to maximize public health safety”; Line 100-102, Page 3: “both traditional and social media platforms have fueled an infodemic in which consumers are oversaturated with information including”), it seems to be inadequate.

3.Line 96-106, Page 3: “Given that those who believe misinformation are less likely to seek out valid guidance and adhere to public health recommendations, we hypothesize that those with little knowledge of COVID-19 will express less worry about contracting the disease.”

This sentence needs to be reconsidered, “believe misinformation” is different with “little knowledge”.

4.Line 107-116, Page 3: What does “News Source” mean in the study?

The authors were asked to define what “News Source” is.

5.Line 111-114, Page 3: “This is likely driven by these networks’ desire to protect the public image of the conservative leadership in power at the time of the pandemic’s onset, as well as to appeal to similarly conservative audiences who value personal liberties over wide-reaching government regulations.”

Actually, I did not see any necessity and sensible for this sentence, it has nothing to do with the research topic of the study, and it is suggested to be deleted.

6.Line 134-135, Page 4: “Survey data was collected in July and August 2020 from 1,219 Amazon MTurk workers……”

The authors were asked to show the basis of sample selection. Only selected “workers”, the study samples seem to be low heterogeneity.

7.Line 183-184, Page 5: “education level (high school degree [ref.] vs. college degree or greater)”

The authors were asked to show the basis of sample selection. Respondents seem to have a relatively high level of education, and none of the respondents have an education level below high school degree, which may affect the results of the study.

8.Line 184-185, Page 5: “employment status (unemployed [ref.], fulltime employment, part-time employment)”

The sample of the study was Amazon MTurk workers, why is there unemployed respondents in “employment status”. The authors were asked to clarify it.

9.The authors divided “COVID-19 Worry” into “Not at all”, “Somewhat”, and “Very” to analysis the associations between MTurk worker characteristics and COVID-19 worry in Table1 and Table2. However, “COVID-19 Worry” was divided into “Not at all or somewhat” and “Very” to do the logistic regression in Table3 and Table4. Why

the division of “COVID-19 Worry” is inconsistent.

10.Line 207-208, Page 5: “Fox News watchers, on the other hand, were less worried (p = 0.016).”

The authors proposed that those who consumed Fox News were less worried about contracting COVID-19, only because of the p-value (p = 0.016)? It seems to be unreasonable and unconvincing.

11.Line 272-274, Page 10: “The fact that these two sets of findings from New York State and the national sample are similar shows that the factors associated with being very worried about contracting COVID-19 are replicable and generalizable across geographic contexts.”

Please authors explain the foundation of the conclusion of “replicable and generalizable across geographic contexts”, the current evidence is insufficient and not rigorous enough.

12.Line 275-277, Page 10: “The association between endorsement of COVID-19 stigmatizing restrictions and worry about contracting the disease is consistent with recent literature on COVID-19 stigma and the notion that fear of infection and stigmatization are closely related.”

Please authors explain the foundation of the conclusion, “recent literature” refer to?

13.Line 282-284, Page 10: “These findings from outside the U.S. appear to operate in contrast to our presumption of a simple relationship driven by the Health Belief Model between COVID-19 stigma, worry, and the uptake of protective health behaviors.”

In this section, the authors were asked to make the explanation why endorsement of COVID-19 stigmatizing restrictions will bring higher worry about contracting COVID-19.

14.Line 313-314, Page 11: “Our findings also demonstrate the importance of preferred news source in shaping levels of COVID-19 worry and potentially the uptake of protective measures thereafter.”

Judging from the results, the study only indicated preferred news source in shaping levels of COVID-19 worry, did not show that news source affects protective measures thereafter.

15.Line 336-338, Page 11: “None of these studies took place in the U.S. context, which may explain why these factors were not significant in the cultural and social environments of our study participants.”

Please authors explain the foundation of the conclusion, so sociodemographic factors were not significant was because the study countries were different?

Author Response

  • First, in general, the description of the Health Belief Model was unclear, please make efforts to revise this section. For instance, the authors proposed that “The Health Belief Model is a widely used conceptual framework to explain the uptake of health behaviors……” (Line 38-39, Page 1), the description of the Health Belief Model was not sufficient, and more literature citations are necessary. In addition,the rationality of the definition of the concepts of perceived susceptibility, perceived severity and perceived threat needs to be supplemented (“Perceived susceptibility is the likelihood that……in question.” Line 42-47, Page 1-2). Thirdly, perceived benefits, perceived barriers and self-efficacy seems to be redundant (Figure1, Page 2), have no relevant to the study and need to be revised. Furthermore, based on the Health Belief Model, how behavioral of COVID-19 worry, COVID-19 stigmatizing restrictions, COVID-19 knowledge, news source and depression fit this Model? Please give further explanation.

Thank you very much for this suggestion. We have made the following revisions:

  1. We more explicitly defined perceived susceptibility and perceived severity with direct quotations from Champion & Skinner 2008.
  2. We clarified that perceived threat is a direct combination of perceived susceptibility and perceived severity.
    1. We simplified the figure of the Health Belief model to reflect the relationship more clearly between perceived susceptibility, perceived severity, and perceived threat.
  3. We deleted the sentence referring to the other constructs of the Health Belief Model and revised the figure to only reflect those constructs relevant to the paper (i.e., we deleted constructs of perceived benefits, perceived barriers, and self-efficacy).

Finally, we also clarify that our measure of COVID-19 worry (our main predictive variable of interest) is a proxy measure for perceived threat. We had considered asking respondents directly in the survey whether they felt threatened by COVID-19. However, we felt that would have been a poor question and instead opted for querying about perceived threat via “degree of worry.”

  • Second, the value and significance of this study was not clear, the conclusions of the study were obvious, new addition to the knowledge base needs to be clear. As the findings stand, little new information is graspable. That is, the impacts of the COVID-19 stigmatizing restrictions, COVID-19 knowledge, news source and depression on behavioral of COVID-19 worry are too obvious. There is simply no surprise with the findings. Great crafting is necessary to articulate the value of work and pinpoint new knowledge created.

We appreciate this opportunity to further strengthen our arguments and now more clearly state that our manuscript makes the following contributions to the literature on COVID-19:

  1. Very few researchers have directly studied the concept of worry in the context of COVID-19. Worry about infectious disease is an important concept to study because it can be associated with positive outcomes, such as the uptake of protective health measures, but also negative outcomes, such as the potential endorsement of stigmatizing beliefs and poor psychological wellbeing (e.g., depression).
  2. Considering this, our study is one of the first to focus on the behavioral correlates of COVID-19 worry. While we did hypothesize certain associations at the outset, we had some surprising findings especially as it pertained to news sources and sociodemographic characteristics. For example, consumers of Fox News and social media were not significantly less likely to be worried about COVID-19, and we did not find any significant differences in levels of worry based on gender, age, race/ethnicity, education level, or employment status.
  3. Of note was the substantial overall prevalence of worry in both samples; we found that 36% of New York State respondents and 45% of national respondents were very worried about COVID-19. This is now in the revised manuscript on Page 11, Lines 295-297.
  4. We found that being very worried about COVID-19 was significantly associated with negative consequences of probable depression and endorsement of stigmatizing restrictions.
  5. Levels of COVID-19 worry were also associated with other correlates of interest, including consumption of commercial news and The New York Times, and COVID-19 knowledge levels.
  6. Our results also have important public health implications; while greater education by public health officials about COVID-19 could lead to greater levels of worry and wider uptake of preventive behaviors, there also needs to be a realization of potential negative effects such as increased levels of stigma and poor psychological wellbeing. This is now in the revised manuscript on Page 13, Lines 408-411.
  • Third, in general, the rationale for conducting this study was unclear, please make efforts to revise the introduction section. And in the literature, please provide more evidence to address the purpose of this study is critical.

We are glad to have the opportunity to further clarify these points and have made the following revisions to the introduction to address this concern:

  1. We have cited the few studies that have assessed predictors of COVID-19 worry in the U.S. context on Page 3, Line 79, including Barber & Kim 2020, Maxfield & Pituch 2021, and Taylor et al. 2020; however, these studies are limited in that they do not address stigma or preferred news source, which we have made sure to note in the Introduction on Page 3, Lines 82-84.
  2. We have added the following sentences in the Introduction to further rationalize the study:
    1. Page 3, Lines 91-94: “It is important to determine these behavioral and psychological correlates in order to frame further examination of COVID-19 worry as a precursor to either constructive or harmful behavior.”
    2. Page 3, Lines 105-109: “Stigmatization leads to psychological, social, economic, and sometimes physical harm to those who are stigmatized with few discrete benefits of reducing disease transmission. It is therefore important to study the stigmatization of COVID-19 due to the very real possibility of these harms, which have materialized in discrimination and harassment toward those of Chinese descent.” (citing Litam & Oh 2020 and Cheah et al. 2020)

  • Fourth, there are few explanations of the rationality for the materials and methods. For instance, how “COVID-19 Knowledge” is measured was not mentioned in the study, “COVID-19 Knowledge” was divided into “A great deal” and “Some/not much” (Table1, Page 5-6). What kind of respondents were defined as "A great deal" and what kind of respondents were defined as "Some/not much", which needs to be supplemented.

We added the following sentences on Page 5, Lines 200-203: COVID-19 knowledge was based on respondents’ self-reported response to the question, “How much have you heard about COVID-19?” with options being ‘a great deal,’ ‘some,’ or ‘not much.’ For purposes of this analysis, COVID-19 knowledge was dichotomized as a great deal (ref.) versus some or not much.

  • In addition, how “Depression” was measured? “Depression” was divided into “No Probable Depression” and “Probable Depression” (Table1, Page 5-6), it should be described in detail.

The measurement of probable depression was edited to read as follows on Page 5, Lines 213-215: “Probable current depression (none [ref.] vs. probable) was based on the widely validated Patient Health Questionnaire, where probable depression was based on an established cut-off score of five or greater.”

  • Furthermore, the authors were asked to clarify the causal relationship between “depression” and “worry of contracting COVID-19”, it is very necessary. For example, there are contradictory statements in the study. “Several of these studies also noted greater perceived risk of contracting COVID-19 to be a significant predictor of depression” (Line 120-122, Page 3). This sentence shows that “perceived risk of contracting COVID-19” affect “depression”. “We therefore hypothesize that, similar to these other studies, respondents with probable depression will express greater worry of contracting COVID-19” (Line 122-123, Page 3). However, this sentence shows that “depression” affect “worry of contracting COVID-19”.

We appreciate the opportunity to clarify this important point and have now removed language implying causality between these two variables. On Page 4, Lines 143-144, we revised the sentence to read, “Several of these studies also noted greater perceived risk of contracting COVID-19 to be significantly associated with depression.” The cross-sectional nature of our study precludes the ability to determine a causal relationship between COVID-19 worry and probable depression, a limitation that we address on Page 12, Lines 385-388.

  • Last but not least, it is noted that the manuscript needs careful editing by someone with expertise in technical English editing paying particular attention to English grammar, spelling, and sentence structure so that the goals and results of the study are clear to the reader.

We appreciate this feedback and our three senior authors (D.D., L.Y., V.C.), who are all native English speakers and established scientists, have read through and edited the manuscript closely.

Specific comments:

  • Line 96-106, Page 3: What does “COVID-19 Knowledge” mean in the study? The authors were asked to define what “COVID-19 Knowledge” is.

We added the following sentences on Page 5, Lines 200-203: COVID-19 knowledge was based on respondents’ self-reported response to the question, “How much have you heard about COVID-19?” with options being ‘a great deal,’ ‘some,’ or ‘not much.’ For purposes of this analysis, COVID-19 knowledge was dichotomized as a great deal (ref.) versus some or not much.

  • Line 96-106, Page 3: In this section, the author needs to explain the relationship between “COVID-19 Knowledge” and “behavior of COVID-19 worry”, however, the authors was emphasizing the role of social media in public health safety (Line 98-100, Page 3: “the importance of technology and social media in keeping the public informed about the latest developments in the COVID-19 pandemic and updated guidelines to maximize public health safety”; Line 100-102, Page 3: “both traditional and social media platforms have fueled an infodemic in which consumers are oversaturated with information including”), it seems to be inadequate.

We have clarified the relationship between COVID-19 knowledge and COVID-19 worry on Pages 3-4, Lines 113-125 as follows: “In a joint statement, the World Health Organization and other multinational agencies acknowledged the importance of technology and media platforms to increasing public knowledge by informing consumers about the latest developments in the COVID-19 pandemic. Given that greater COVID-19 news consumption at the start of the pandemic was shown to be associated with anticipated mental health challenges (e.g., depression) (citing Piltch-Loeb, Merdjanoff, & Meltzer 2021), we hypothesize that those with greater knowledge of COVID-19 will express more worry about contracting the disease.”

We have, furthermore, removed the sentence that read: “However, both traditional and social media platforms have fueled an infodemic in which consumers are oversaturated with information including “deliberate attempts to disseminate wrong information to undermine the public health response and advance alternative agendas of groups or individuals.”

  • Line 96-106, Page 3: “Given that those who believe misinformation are less likely to seek out valid guidance and adhere to public health recommendations, we hypothesize that those with little knowledge of COVID-19 will express less worry about contracting the disease.” This sentence needs to be reconsidered, “believe misinformation” is different with “little knowledge”.

This language has been removed from the manuscript.

  • Line 107-116, Page 3: What does “News Source” mean in the study? The authors were asked to define what “News Source” is.

We added the following sentence on Page 5, Lines 203-204: Respondents were also asked, “From which of these sources do you regularly get your news from?”

  • Line 111-114, Page 3: “This is likely driven by these networks’ desire to protect the public image of the conservative leadership in power at the time of the pandemic’s onset, as well as to appeal to similarly conservative audiences who value personal liberties over wide-reaching government regulations.” Actually, I did not see any necessity and sensible for this sentence, it has nothing to do with the research topic of the study, and it is suggested to be deleted.

We have deleted this sentence per the reviewer’s suggestion.

  • Line 134-135, Page 4: “Survey data was collected in July and August 2020 from 1,219 Amazon MTurk workers……” The authors were asked to show the basis of sample selection. Only selected “workers”, the study samples seem to be low heterogeneity.

The sample’s potential low heterogeneity is addressed in the Limitations section on Page 13, Lines 391-394.

  • Line 183-184, Page 5: “education level (high school degree [ref.] vs. college degree or greater)” The authors were asked to show the basis of sample selection. Respondents seem to have a relatively high level of education, and none of the respondents have an education level below high school degree, which may affect the results of the study.

This is addressed in the Limitations section on Page 13, Lines 393-396.

  • Line 184-185, Page 5: “employment status (unemployed [ref.], fulltime employment, part-time employment)” The sample of the study was Amazon MTurk workers, why is there unemployed respondents in “employment status”. The authors were asked to clarify it.

Amazon MTurk workers are not employed by Amazon; rather, they provide this service through an Amazon platform hosted crowdsourcing online marketplace on a contractual basis for supplemental income. That is, respondents can be unemployed while also participating in Amazon MTurk for supplemental income (respondents can be paid as little as 50 cents to one USD per survey based on time spent). 

  • The authors divided “COVID-19 Worry” into “Not at all”, “Somewhat”, and “Very” to analysis the associations between MTurk worker characteristics and COVID-19 worry in Table1 and Table2. However, “COVID-19 Worry” was divided into “Not at all or somewhat” and “Very” to do the logistic regression in Table3 and Table4. Why the division of “COVID-19 Worry” is inconsistent.

This is addressed on Page 5, Lines 219-223, which states: “Following the Health Belief Model’s presumption that those who were ‘very worried’ would be most likely to take preventive health measures and adhere to public health guidelines, we dichotomized COVID-19 into ‘not at all or somewhat’ (ref.) versus ‘very’ worried for multivariable analyses.”

  • Line 207-208, Page 5: “Fox News watchers, on the other hand, were less worried (p = 0.016).” The authors proposed that those who consumed Fox News were less worried about contracting COVID-19, only because of the p-value (p = 0.016)? It seems to be unreasonable and unconvincing.

We have revised the sentence to read: “Watching Fox News was associated with less worry (p = 0.016).”

  • Line 272-274, Page 10: “The fact that these two sets of findings from New York State and the national sample are similar shows that the factors associated with being very worried about contracting COVID-19 are replicable and generalizable across geographic contexts.” Please authors explain the foundation of the conclusion of “replicable and generalizable across geographic contexts”, the current evidence is insufficient and not rigorous enough.

We revised the sentence on Page 11, Lines 306-310 to read: “The fact that these two sets of findings from New York State and the national sample are similar shows that the factors associated with being very worried about contracting COVID-19 are replicable and potentially generalizable from the single state to a nationally-based sample, at least in the U.S.

  • Line 275-277, Page 10: “The association between endorsement of COVID-19 stigmatizing restrictions and worry about contracting the disease is consistent with recent literature on COVID-19 stigma and the notion that fear of infection and stigmatization are closely related.” Please authors explain the foundation of the conclusion, “recent literature” refer to?

We revised the sentence on Page 11, Lines 311-315 to read: “The association between endorsement of COVID-19 stigmatizing restrictions and worry about contracting the disease is consistent with recent studies showing the negative impact of COVID-19 stigma on taking recommended public health actions, and the notion that stigmatization and fear of infection are closely related.”

  • Line 282-284, Page 10: “These findings from outside the U.S. appear to operate in contrast to our presumption of a simple relationship driven by the Health Belief Model between COVID-19 stigma, worry, and the uptake of protective health behaviors.” In this section, the authors were asked to make the explanation why endorsement of COVID-19 stigmatizing restrictions will bring higher worry about contracting COVID-19.

We edited the sentence on Page 11, Lines 319-322 to read: “These findings from outside the U.S. appear to operate in contrast to a simple relationship between COVID-19 stigma, worry, and the uptake of protective health behaviors.”

  • Line 313-314, Page 11: “Our findings also demonstrate the importance of preferred news source in shaping levels of COVID-19 worry and potentially the uptake of protective measures thereafter.” Judging from the results, the study only indicated preferred news source in shaping levels of COVID-19 worry, did not show that news source affects protective measures thereafter.

We revised the sentence on Page 12, Lines 352-353 to read: “Our findings also demonstrate the importance of the association between preferred news source and levels of COVID-19 worry.”

  • Line 336-338, Page 11: “None of these studies took place in the U.S. context, which may explain why these factors were not significant in the cultural and social environments of our study participants.” Please authors explain the foundation of the conclusion, so sociodemographic factors were not significant was because the study countries were different?

We revised the sentence on Page 12, Lines 381-383 to read: “The sociodemographic factors shown to be significant in our study may not have showed differences in these other studies due to differing country contexts.”

Reviewer 2 Report

I read with great interest the Manuscript titled “Behavioral correlates of COVID-19 worry: stigma, knowledge, and news source” (ijerph-1435765), which falls within the aims of International Journal of Environmental Research and Public Health.     

In my honest opinion, the topic is interesting and worthy of attention. Moreover, the methodology is accurate, and the conclusions are supported by the data analysis. Nevertheless, the authors should clarify some points and improve the discussion by citing relevant and novel key articles about the topic.

Authors should consider the following recommendations:

  • Did the authors calculate the sample size before starting their study? Otherwise, the authors should perform a post-hoc calculation of the power of their study to ensure the adequate significance of their results.
  • The authors correctly underlined that the constructs of the Health Belief Model are influenced by a complex interplay of sociodemographic, cultural, psychological, ideological, and structural variables. it would be appropriate to cite some references on this topic. Some interesting articles to mention are: DOI: 10.1111/j.1539-6924.2006.00867.x; DOI: 10.1080/10810730.2014.989342
  • It would be interesting to mention, even briefly, the different impacts of the COVID-19 pandemic on the quality of life and mental health of specific populations such as students or pregnant women. Some recent articles on this topic are the following: doi:10.3390/ijerph17165933; doi: 10.3389/fpsyg.2020.559951.

Author Response

1. Did the authors calculate the sample size before starting their study? Otherwise, the authors should perform a post-hoc calculation of the power of their study to ensure the adequate significance of their results.

Post-hoc power calculations revealed that our study had a power of 0.98 for both the New York State and national samples to detect an odds ratio of 2.44 at the alpha = 0.05 level.

2. The authors correctly underlined that the constructs of the Health Belief Model are influenced by a complex interplay of sociodemographic, cultural, psychological, ideological, and structural variables. it would be appropriate to cite some references on this topic. Some interesting articles to mention are: DOI: 10.1111/j.1539-6924.2006.00867.x; DOI: 10.1080/10810730.2014.989342

We have cited the studies you recommended, as well as Palmer 2010 and Renn & Rohrmann 2000.

3. It would be interesting to mention, even briefly, the different impacts of the COVID-19 pandemic on the quality of life and mental health of specific populations such as students or pregnant women. Some recent articles on this topic are the following: doi:10.3390/ijerph17165933; doi: 10.3389/fpsyg.2020.559951.

We have added a sentence on page 12, Lines 372-375 that reads as follows: “This is especially the case for certain subpopulations that have disproportionately experienced poor mental health and reduced quality of life as a result of the COVID-19 pandemic, including students, pregnant women, frontline healthcare workers, and East Asians who have experienced verbal and physical harassment” citing the studies you suggested and Ceri & Cicek 2020 and Wu et al. 2020.

Reviewer 3 Report

congratulations to the authors for the well-written article. Please address the few comments

  1. The study was done during the early phase of the pandemic. There was not a clear understanding of the disease, can it play a significant role in the response from the Survey participants?
  2. Did the authors identify if the participants were from urban or rural areas? Does it make in difference in the outcome?
  3. The early response from the federal government was not clear during the early phase of the pandemic, which may have had some impact on the participants.
  4. What do the authors think about the COVID 19 News fatigue in people's behavior?

Author Response

1. The study was done during the early phase of the pandemic. There was not a clear understanding of the disease, can it play a significant role in the response from the Survey participants?

We have added a sentence on Page 11, Lines 304-306 that reads: “These findings may reflect the early phase of the pandemic during which most people did not have a clear understanding of COVID-19 and the federal government failed to use an evidence-based approach in its response.”

2. Did the authors identify if the participants were from urban or rural areas? Does it make in difference in the outcome?

We have included this in the Limitations section on Page 13, Lines 393-396: “The respondents in our study had higher education levels than the general population, and it was also not possible to discern whether they came from urban or rural areas, which may have also influenced their responses.

3. The early response from the federal government was not clear during the early phase of the pandemic, which may have had some impact on the participants.

We have added a sentence on Page 11, Lines 304-306 that reads: “These findings may reflect the early phase of the pandemic during which most people did not have a clear understanding of COVID-19 and the federal government failed to use an evidence-based approach in its response.” It cites Solinas-Saunders 2020.

4. What do the authors think about the COVID 19 News fatigue in people's behavior?

This is a very interesting question and one that we will explore in further iterations of our national surveys.